# Clinical Utility of the Tokyo Guidelines 2018 for Acute Cholangitis in the Emergency Department and Comparison with Novel Markers (Neutrophil-to-Lymphocyte and Blood Nitrogen Urea-to-Albumin Ratios)

**DOI:** 10.3390/jcm13082306

**Published:** 2024-04-16

**Authors:** Hyun-Min Jung, Jinhui Paik, Minsik Lee, Yong Won Kim, Tae-Youn Kim

**Affiliations:** 1Department of Emergency Medicine, Inha University Hospital, College of Medicine, Inha University, 27, Inhang-ro, Jung-gu, Incheon 22332, Republic of Korea; hyunmin72@hanmail.net (H.-M.J.); riven2ne@naver.com (J.P.); 2Department of Emergency Medicine, Dongguk University Ilsan Hospital, College of Medicine, Dongguk University, Goyang 10326, Republic of Korea; hrabbit33@naver.com (M.L.); chiefong@naver.com (Y.W.K.)

**Keywords:** cholangitis, Tokyo guidelines, clinical outcome, novel biomarker

## Abstract

**Introduction**: The Tokyo Guidelines 2018 (TG2018) is a scoring system used to recommend the clinical management of AC. However, such a scoring system must incorporate a variety of clinical outcomes of acute cholangitis (AC). In an emergency department (ED)-based setting, where efficiency and practicality are highly desired, clinicians may find the application of various parameters challenging. The neutrophil-to-lymphocyte ratio (NLR) and blood urea nitrogen-to-albumin ratio (BAR) are relatively common biomarkers used to assess disease severity. This study evaluated the potential value of TG2018 scores measured in an ED to predict a variety of clinical outcomes. Furthermore, the study also compared TG2018 scores with NLR and BAR scores to demonstrate their usefulness. **Methods**: This retrospective observational study was performed in an ED. In total, 502 patients with AC visited the ED between January 2016 and December 2021. The primary endpoint was to evaluate whether the TG2018 scoring system measured in the ED was a predictor of intensive care, long-term hospital stays (≥14 days), percutaneous transhepatic biliary drainage (PTBD) during admission care, and endotracheal intubation (ETI). **Results**: The analysis included 81 patients requiring intensive care, 111 requiring long-term hospital stays (≥14 days), 49 requiring PTBD during hospitalization, and 14 requiring ETI during hospitalization. For the TG2018 score, the adjusted OR (aOR) using (1) as a reference was 23.169 (95% CI: 9.788–54.844) for (3) compared to (1). The AUC of the TG2018 for the need for intensive care was 0.850 (95% CI: 0.815–0.881) with a cutoff of >2. The AUC for long-term hospital stays did not exceed 0.7 for any of the markers. the AUC for PTBD also did not exceed 0.7 for any of the markers. The AUC for ETI was the highest for BAR at 0.870 (95% CI: 0.837–0.899) with a cutoff value of >5.2. **Conclusions**: The TG2018 score measured in the ED helps predict various clinical outcomes of AC. Other novel markers such as BAR and NLR are also associated, but their explanatory power is weak.

## 1. Introduction

Acute cholangitis (AC) is a common condition observed in the emergency department (ED) in patients presenting with abdominal pain; proper diagnosis and treatment are essential [1]. The course of AC varies among patients, with some developing serious complications such as biliary sepsis [2]. Therefore, early recognition of the clinical outcomes of AC is important for clinicians working in the ED when planning treatment [3]. The Tokyo Guidelines 2018 (TG2018) is a scoring system used to recommend the clinical management of AC [4]. Moreover, the system predicts the severity and prognosis of AC based on a patient’s clinical characteristics and laboratory tests [4,5]. The early use of such a scoring system, especially in the ED, can provide important information for initial diagnosis and treatment planning [6]. These international guidelines grade AC based on 11 parameters after diagnosing AC using components A (systemic inflammation), B (cholestasis), and C (imaging) [4]. Grade I involves antibiotics and general supportive care, and biliary drainage was considered if no response to the initial treatment was observed; grade II encompasses early endoscopic or percutaneous transhepatic biliary drainage (PTBD) in addition to antibiotics and general supportive care. Grade III includes urgent biliary drainage and respiratory/circulatory management [4]. In addition, TG18 preceding guideline TG13 demonstrated that the 30-day mortality rates of patients with grades III, II, and I were 5.1%, 2.6%, and 1.2%, respectively, and increased significantly with disease severity [7]. However, such a scoring system must incorporate a variety of clinical outcomes for AC. In an ED-based setting, where efficiency and practicality are highly desired, clinicians may find the application of various parameters challenging [8]. Generally, emergency physicians are essential for providing quick and effective care in a variety of emergencies [8,9]. However, limited resources must be utilized efficiently to provide optimal care in emergency settings [10]. Within the ED, the decision to admit a patient to the intensive care unit is a critical process that necessitates considering the severity of the patient’s condition, the effectiveness of the treatment, and available resources that directly impact the quality and efficiency of patient care [11].

The neutrophil-to-lymphocyte ratio (NLR) and blood urea nitrogen-to-albumin ratio (BAR) are relatively common biomarkers used to assess disease severity [12,13,14]. These biomarkers reflect changes that occur during inflammation, infection, and various types of stress responses [14,15]. Previous studies have demonstrated that the NLR is useful for tracking the clinical course of AC, but no studies have compared it with TG2018 [16]. This study evaluated the potential value of TG2018 scores measured in the ED to predict a variety of clinical outcomes, including whether AC requires intensive care unit treatment. Furthermore, the study also compared TG2018 scores with NLR and BAR scores to demonstrate their usefulness.

## 2. Methods

### 2.1. Study Design and Participants

This retrospective observational study was performed in the ED of a tertiary university hospital that receives 35,000 patients annually, with approximately 100 patients presenting with AC. This study was approved by the Institutional Review Board of Dongguk University Ilsan Hospital (IRB No. DUIH 2021-10-016-002). The study protocol conformed with the ethical guidelines of the 1975 Declaration of Helsinki.

As the study involved retrospective and observational analyses, the requirement for informed consent was waived, and patient records and information were anonymized before analysis. Computerized hospital records were reviewed, and patients for whom “acute cholangitis (K83.0)” was used as a discharge code based on the International Classification of Diseases, 10th revision coding, were initially considered for study selection.

In total, 502 patients with AC visited the ED between January 2016 and December 2021. The exclusion criteria were as follows: (1) patients who did not undergo serum laboratory tests and (2) those with age <19 years.

### 2.2. Study Variables

Data were retrospectively collected from electronic medical records. These included age, sex, comorbidities, initial presenting blood pressure, heart rate, hemoglobin level, white blood cell count, neutrophil count, lymphocyte count, platelet count, albumin, total bilirubin, prothrombin time–international normalized ratio, blood urea nitrogen, creatinine, and C-reactive protein (CRP) levels measured within 1 h of ED arrival. The comorbidities included hypertension, diabetes mellitus, liver cirrhosis, and chronic renal failure. Additionally, the BAR and NLR were measured in the ED using serum laboratory parameters.

### 2.3. Study Endpoints

The primary endpoint was to evaluate whether the TG2018 scoring system measured in the ED was a predictor of intensive care, long-term hospital stays (≥14 days), PTBD during admission care, and endotracheal intubation (ETI). If the TG2018 system served as a predictor, the optimal cutoff levels predicting intensive care, long-term hospital stay, PTBD, and ETI could be determined. The secondary endpoint was to evaluate whether the NLR and BAR measured in the emergency department are predictors of clinical outcomes. Predictive performance was compared to the TG2018.

Intensive unit admission was determined based on clinician judgment, with vital signs serving as objective parameters, along with clinical examination and acute-onset physical signs when necessary. We categorized patients using a priority model [17]. Priority 1 was considered for patients requiring intensive care and monitoring, such as patients on a ventilator or intravenous cardiovascular medications, those with respiratory failure requiring ventilation after surgery, and those requiring invasive monitoring. Priority 2 was assigned to patients who might require immediate treatment at any time during intensive monitoring and those with chronic diseases that can deteriorate rapidly. Priority 3 was assigned to patients with underlying or acute diseases that may require intensive care. Long-term hospital stay was defined as hospitalization lasting longer than 14 days [18]. The indications for PTBD were as follows: (1) anatomical reasons—the patient’s biliary structures were in an unusual position or had been altered by previous surgery, making endoscopic retrograde cholangiopancreatography (ERCP) difficult, and (2) failure of ERCP, the procedure was attempted and did not successfully eliminate the cause. The typical indications for ETI include (1) respiratory failure, (2) loss of consciousness, or (3) airway obstruction [6].

### 2.4. Statistical Analysis

Categorical variables are presented as frequencies and percentages, whereas continuous variables are presented as means and standard deviations or as medians and interquartile ranges. Chi square or Fisher’s exact tests were used to compare categorical variables, whereas two-sample *t*-tests or Mann–Whitney U-tests were used to compare continuous variables. Normality was first assessed using the Shapiro–Wilk test. Logistic regression analysis was used to identify factors that predicted intensive care, long-term hospital stays, PTBD, and ETI. The results are expressed as odds ratios (ORs) with 95% confidence intervals (CIs). The area under the receiver operating characteristic curve was used to evaluate the ability of TG2018, BAR, and NLR to predict clinical outcomes. In addition, the area under the curve (AUC) was used to compare the predictive ability of each method. Statistical significance was set at *p* < 0.05, and all statistical analyses were conducted using IBM SPSS Statistics for Windows (version 23.0; IBM, Armonk, NY, USA) and MedCalc Statistical software version 17.5.3 (MedCalc Software, Ostend, Belgium).

## 3. Results

### 3.1. General Characteristics

In total, 502 consecutive patients with AC were enrolled in this study. Twenty patients were excluded due to the absence of serum laboratory tests. Finally, 482 patients with AC were included (Figure 1).

Table 1 displays the baseline patient characteristics. Of the 482 patients analyzed, 248 (51.5%) were men. General characteristics, including comorbidities, vital signs, and laboratory test results, are demonstrated in Table 1. The median hospital stay was 8 days (1–147). The median TG2018 score was 1 (1–3). The median BAR was 3.84 (1–111.11), and the NLR was 9.91 (0.67–194.20).

### 3.2. Comparison of Clinical Outcomes

The analysis included 81 patients requiring intensive care, 111 requiring long-term hospital stays (≥14 days), 49 requiring PTBD during hospitalization, and 14 requiring ETI during hospitalization. The patients requiring intensive care had significantly higher BARs, NLRs, CRP levels, and TG2018 scores than those who did not require intensive care. Additionally, the patients requiring long-term hospital stays also had high levels of BARs, CRP, and TG2018 scores. The patients requiring PTBD and ETIs also exhibited significantly elevated BARs, NLRs, CRP levels, and TG2018 scores (Table 2). Overall, 325 patients underwent ERCP, and the TG2018 scores and ERCP rates were not statistically significant (Appendix A). The results for each outcome by TG2018 grade are presented in Appendix A. The BAR, NLR, and CRP levels by TG2018 grade are presented, and there were statistically significant differences as the severity of AC increased (Appendix A).

### 3.3. Factors Influencing Clinical Outcomes

Multilogistic regression analysis was performed to identify factors associated with intensive care. For the TG2018 score, the adjusted OR (aOR) using (1) as a reference was 23.169 (95% CI: 9.788–54.844) for (3) compared to (1), although no association was observed with a score of (2). The aOR for the BAR was 1.124 (95% CI: 1.042–1.213), for the NLR, it was 1.027 (95% CI: 1.010–1.045), and for CRP, it was 1.041 (95% CI: 1.002–1.081). The aOR of the TG2018 score for PTBD was 2.434 (95% CI: 1.122–5.284) with a score of (3), compared to a score of (1), with no association with the (2) group. The BAR and NLR were not associated with long-term hospital stays. The aOR of the TG2018 score for long-term hospital stay was 3.267 (95% CI: 1.794–5.949) with a score of (3), compared to a score of (1), with no association with the (2) group. The aOR of the CRP level for a long-term hospital stay was 1.040 (1.005–1.077). The BAR, NLR, and CRP levels were not associated with PTBD. In the context of requiring ETI, the TG2018 score demonstrated an association with an aOR of 16.146 (95% CI: 1.823–143.028) for a score of (3) when a score of (1) was used as a reference. No patient had a score of (2). The BAR, NLR, and CRP levels were not associated with the need for ETI (Table 3).

### 3.4. Comparison of Accuracy of Variables for Predicting Outcomes

Table 4 presents the accuracy and cutoff values for each biomarker and the clinical outcomes of the TG2018. The TG2018 scoring system exhibited the highest AUC for predicting the need for intensive care. The AUC of the TG2018 for the need for intensive care was 0.850 (95% CI: 0.815–0.881) with a cutoff of >2. The next highest was the BAR at 0.765 (95% CI: 0.724–0.802). The AUC for long-term hospital stays did not exceed 0.7 for any of the markers. The AUC of the TG2018 for predicting long-term hospital stay was the highest at 0.680 (95% CI: 0.636–0.721) with a cutoff value of >1. Moreover, the AUC for PTBD also did not exceed 0.7 for any of the markers. However, in comparison to the other markers, the AUC was higher with a BAR of 0.651 (95% CI: 0.607–0.694). The AUC for ETI was the highest for the BAR at 0.870 (95% CI: 0.837–0.899) with a cutoff value of >5.2. The TG2018 was 0.854 (95% CI: 0.819–0.884) with a cutoff value of >2 (Table 4 and Figure 2).

## 4. Discussion

In this study, the TG2018 measured in the ED was a useful predictor of intensive care, long-term hospital stay, PTBD, and ETI in patients with AC. In addition to the TG2018 score, the BAR, NLR, and CRP levels were also associated with intensive care. Furthermore, the TG2018 score, BAR, and CRP levels were associated with long-term hospital stays. For PTBD and ETI, TG2018 was the only factor with a score of 3. In particular, the AUC for intensive care and ETI was greater than 0.8, indicating that it was a good predictive marker. Additionally, the BAR and NLR, which are relatively novel markers, were identified to be fair predictors of intensive care and ETI, with an AUC of >0.7. However, the BAR and NLR were not associated with ETI in the multilogistic regression analysis, making them difficult for clinicians to apply.

To the best of our knowledge, this is the first study to present practical ED factors for the clinical outcomes of AC. The existing TG2018 guidelines categorize AC into mild, moderate, and severe and provide algorithms for supportive care, early ERCP biliary drainage, and emergent ERCP within 6 h [6]. In particular, in severe forms of cholangitis, if ERCP fails, endoscopic ultrasound-guided choledochoduodenostomy is performed, and if that fails, PTBD is performed [19]. 

As the first study to apply a validated scoring system to different clinical outcomes, we demonstrated its utility by comparing the system to relatively new markers such as the BAR and NLR. The BAR is a particularly useful marker for assessing renal function and nutritional status [20]. Additionally, the BAR has been applied to clinical outcomes in critically ill patients and has proven to be useful in EDs, especially for conditions such as sepsis and upper gastrointestinal bleeding [21,22,23]. The NLR is primarily a biomarker of inflammation and infection [15]. Specifically, the NLR has proven its utility as a predictor of early clinical outcomes across diverse clinical conditions such as sepsis, acute phase infection, cancer, and postoperative complications [24,25]. 

Various scoring systems have been developed for intensive care. The most widely used is the Acute Physiology and Chronic Health Evaluation II score, which uses physiological variables, such as age. The Sequential Organ Failure Assessment score also assesses the respiratory, coagulation, hepatic, cardiac, and central nervous systems, as well as renal function [26]. The Simplified Acute Physiology Score II also predicts the risk of intensive care mortality based on the patient’s physiological status, medical history, and clinical condition [27]. However, for clinicians to use these scoring systems in the ED as markers of clinical deterioration or death in the intensive care unit, owing to the variety of parameters, it is difficult [28]. Outside the operating room, ETI is a high-risk procedure with high morbidity and mortality [29]. When compared to the operating room, ETI is disadvantageous for complex airway management, especially due to the lack of additional equipment or specialized personnel [30]. Therefore, the decision to intubate in intensive care should be made appropriately, considering the different clinical circumstances. The Glasgow Coma Scale score, poor respiratory drive, questionable airway patency, hypoxia, and hypercarbia are commonly considered, and various clinical signs should be explored [31]. Hospital length of stay is an important monitoring metric for hospital resources, staff, and equipment. Methods for predicting a patient’s length of stay should offer comprehensive coverage and encompass various approaches, regardless of the patient’s condition [32]. Therefore, predicting long hospital stays early in EDs is important. To help clinicians with early prediction, this study presents various scoring systems and biomarkers.

## 5. Limitations

This study had several limitations. First, this study is limited by being a retrospective study. There may be missing data during the data collection process, which makes it unrepresentative of the entire population and prone to selection bias. Second, this study was conducted in the emergency center of a single hospital, which may have resulted in a small sample size. However, to reduce the possibility of this bias, we examined all patients with acute biliary tract infections who visited an emergency center in January 2016. Third, this study is not universally applicable as the indications for intensive care and PTBD vary between hospitals. However, we used a priority model, which is used in many hospitals, to determine intensive care. PTBD was determined after ERCP or according to the guidelines, considering the structure of the patient. Well-designed prospective studies are needed to investigate these limitations, and external validation is required. Fourth, due to the retrospective nature of the study, there are many factors that may contribute to an increased length of hospital stay, and we cannot rule out confounding among all of them. In particular, intensive care and long-term hospital stays can be associated not only with cholangitis but also with age and other underlying medical conditions. Long-term hospital stay is not only associated with a clinical condition but also the possibility of discharging patients to step-down or lower-level hospitals, especially among elderly people.

## 6. Conclusions

The TG2018 score measured in the ED helps predict various clinical outcomes for AC. Other novel markers such as the BAR and NLR are also associated, but their explanatory power is weak.

## Figures and Tables

**Figure 1 jcm-13-02306-f001:**
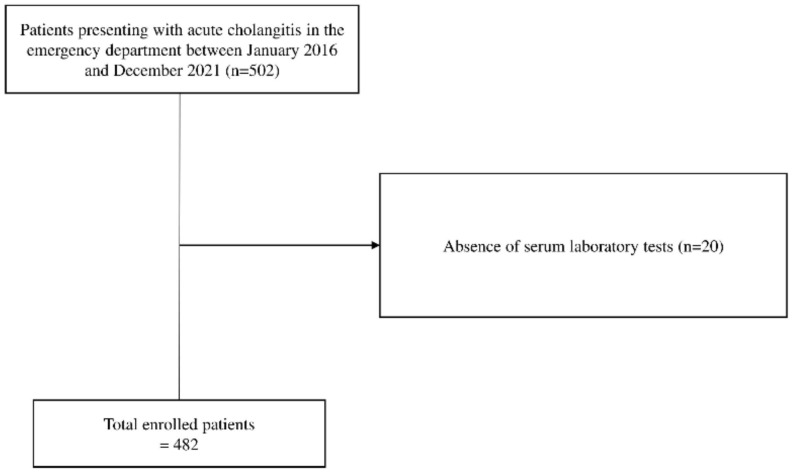
Flowchart of patient screening and selection during the study enrollment.

**Figure 2 jcm-13-02306-f002:**
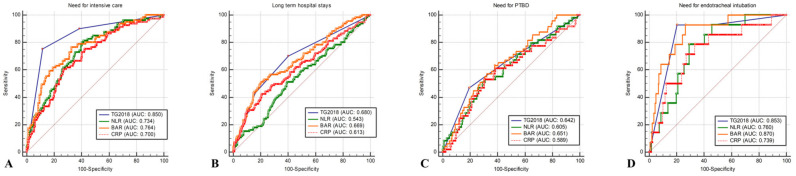
Comparison of accuracy of variables for predicting outcomes. (**A**) Need for intensive care. (**B**) Long-term hospital stays. (**C**) Need for PTBD. (**D**) Need for endotracheal intubation.

**Table 1 jcm-13-02306-t001:** Baseline characteristics of enrolled patients.

	All Patients (n = 482)
Age	75 (23–98)
Male sex, n (%)	248 (51.5%)
Hypertension, n (%)	242 (50.2%)
Diabetes mellitus, n (%)	125 (25.9%)
Liver cirrhosis, n (%)	16 (3.3%)
Chronic renal failure, n (%)	18 (3.7%)
Systolic blood pressure (mmHg)	128 (52–269)
Diastolic blood pressure (mmHg)	71 (40–149)
Heart rate (bpm)	86.5 (20–153)
Hemoglobin level (g/dL)	12.95 (4.6–19.3)
WBC count (×10^9^/L)	9660 (1830–36,000)
Neutrophil count (×10^9^/L)	8490 (3320–9780)
Lymphocyte count (×10^9^/L)	860 (100–5020)
Platelet (×10^9^/L)	193,000 (13,000–523,000)
Albumin (g/dL)	4.1 (1.2–5.2)
Total bilirubin (mg/dL)	2.2 (0.2–23.5)
PT-INR	1.1 (0.86–3.27)
BUN (mg/dL)	15.55 (4.5–200)
Serum lactate dehydrogenase (U/L)	282 (105–2551)
Creatinine (mg/dL)	0.88 (0.33–7.88)
Hospital duration (days)	8 (1–147)
TG2018	1 (1–3)
BAR	3.84 (1–111.11)
NLR	9.91 (0.67–194.20)
C-reactive protein (mg/dL)	3.40 (0.03–31.02)

Abbreviations: WBC; white blood cell, PT-INR; prothrombin time–international normalized ratio, BUN; blood urea nitrogen, TG2018; Tokyo guidelines 2018, BAR; blood urea nitrogen-to-albumin ratio, NLR; neutrophil-to-lymphocyte ratio.

**Table 2 jcm-13-02306-t002:** Comparison of clinical outcomes of patients according to BAR, NLR, CRP, and TG2018.

Factor	Clinical Outcomes		*p*
	Intensive Care		
	No (n = 401)	Yes (n = 81)	
BAR	3.62 (1–35.06)	6.14 (2.18–111.11)	0.000
NLR	8.42 (0.67–86.55)	17.76 (0.95–194.20)	0.000
CRP	2.79 (0.03–30.14)	7.43 (0.03–31.02)	0.000
TG2018	1 (1–3)	3 (1–3)	0.000
	Long-term hospital stays		
	No (n = 371)	Yes (n = 111)	
BAR	3.70 (1.00–111.11)	5.32 (1.66–35.06)	0.000
NLR	9.51 (0.67–194.20)	11.91 (0.95–96.10)	0.168
CRP	2.91 (0.03–31.02)	5.12 (0.04–29.9)	0.000
TG2018	1 (1–3)	2 (1–3)	0.000
	PTBD		
	No (n = 433)	Yes (n = 49)	
BAR	3.75 (1.00–111.11)	5.08 (2.43–24.68)	0.001
NLR	9.55 (0.67–194.20)	14.98 (1.19–96.10)	0.016
CRP	3.27 (0.03–31.02)	6.21 (0.05–24.31)	0.040
TG2018	1 (1–3)	2 (1–3)	0.000
	Endotracheal intubation		
	No (n = 468)	Yes (n = 14)	
BAR	3.79 (1–111.11)	9.24 (3.46–24.37)	0.000
NLR	9.58 (0.67–194.20)	20.16 (5.30–69.86)	0.001
CRP	3.27 (0.03–31.02)	10.70 (0.14–25.30)	0.002
TG2018	1 (1–3)	3 (1–3)	0.000

Abbreviations: PTBD; percutaneous transhepatic bile drainage, BAR; blood urea nitrogen-to-albumin ratio, NLR; neutrophil-to-lymphocyte ratio, CRP; c-reactive protein, TG2018; Tokyo guideline 2018.

**Table 3 jcm-13-02306-t003:** Factors influencing intensive care, PTBD, and endotracheal intubation.

Factor	Need for Intensive Care (n = 81)	Long-Term Hospital Stays (≥14 Days) (n = 111)	Need for PTBD (n = 49)	Need for Endotracheal Intubation (n = 14)
	Adjust OR ^a^ (95% CI)	Adjust OR ^a^ (95% CI)	Adjust OR ^a^ (95% CI)	Adjust OR ^a^ (95% CI)
BAR	1.124 (1.042–1.213)	1.014 (0.974–1.059)	1.001 (0.961–1.044)	1.023 (0.983–1.066)
NLR	1.027 (1.010–1.045)	0.997 (0.984–1.010)	1.008 (0.994–1.023)	1.008 (0.986–1.029)
CRP	1.041 (1.002–1.081)	1.040 (1.005–1.077)	1.006 (0.961–1.053)	1.038 (0.969–1.112)
TG2018 (1)	reference	reference	reference	reference
(2)	2.435 (0.916–6.472)	1.713 (0.934–3.141)	0.837 (0.342–2.045)	-
(3)	23.169 (9.788–54.844)	3.267 (1.794–5.949)	2.434 (1.122–5.284)	16.146 (1.823–143.028)

^a^ Controlled for age, sex, systolic blood pressure, diastolic blood pressure, heart rate, and hemoglobin level. Abbreviations: PTBD; percutaneous transhepatic bile drainage, OR; odds ratio, BAR; blood urea nitrogen-to-albumin ratio, NLR; neutrophil-to-lymphocyte ratio, CRP; c-reactive protein, TG2018; Tokyo guidelines 2018.

**Table 4 jcm-13-02306-t004:** Predictive variable of clinical outcomes (need for intensive care, need for PTBD, and need for endotracheal intubation).

Need for Intensive Care	AUC (95% CI)	Cutoff	Sensitivity (95% CI)	Specificity (95% CI)	PPV (95% CI)	NPV (95% CI)
BAR	0.765 (0.724–0.802)	>4.26	76.54 (65.8–85.2)	67.08 (92.2–71.7)	32.0 (28.1–36.1)	93.4 (90.5–95.5)
NLR	0.734 (0.692–0.773)	>10.54	81.48 (71.3–89.2)	58.60 (53.6–63.5)	28.4 (25.4–31.7)	94.0 (90.8–96.1)
CRP	0.701 (0.658–0.742)	>6.2	60.49 (49.0–71.2)	73.07 (68.4–77.3)	31.2 (26.3–36.5)	90.2 (87.4–92.3)
TG2018	0.850 (0.815–0.881)	>2	75.31 (64.5–84.2)	88.28 (84.7–91.3)	56.5 (49.1–63.6)	94.7 (92.4–96.3)
Long-term hospital stays	AUC (95% CI)	Cutoff	Sensitivity (95% CI)	Specificity (95% CI)	PPV (95% CI)	NPV (95% CI)
BAR	0.669 (0.625–0.711)	>5.26	51.35 (41.7–61.0)	79.51 (75.0–83.5)	42.9 (36.4–49.6)	84.5 (81.8–86.9)
NLR	0.543 (0.497–0.588)	>11.83	51.35 (41.7–61.0)	60.65 (55.5–65.7)	28.1 (23.8–32.7)	80.6 (77.2–83.7)
CRP	0.613 (0.568–0.656)	>7.75	42.34 (33.0–52.1)	78.98 (74.5–83.0)	37.6 (31.0–44.7)	82.1 (79.5–84.4)
TG2018	0.680 (0.636–0.721)	>1	70.27 (60.9–78.6)	59.57 (54.4–64.6)	34.2 (30.4–38.2)	87.0 (83.3–90.0)
Need for PTBD	AUC (95% CI)	Cutoff	Sensitivity (95% CI)	Specificity (95% CI)	PPV (95% CI)	NPV (95% CI)
BAR	0.651 (0.607–0.694)	>4.25	63.27 (48.3–76.6)	62.12 (57.4–66.7)	15.9 (12.9–19.5)	93.7 (91.1–95.6)
NLR	0.605 (0.560–0.649)	>14.19	53.06 (38.3–67.5)	69.05 (64.5–73.4)	16.3 (12.6–20.7)	92.9 (90.6–94.6)
CRP	0.590 (0.544–0.634)	>4.59	61.22 (46.2–74.8)	61.43 (56.7–66.0)	15.2 (12.2–18.8)	93.3 (90.7–95.3)
TG2018	0.642 (0.598–0.685)	>2	46.94 (32.5–61.7)	80.37 (76.3–84.0)	21.3 (16.0–27.8)	93.0 (91.0–94.6)
Need for endotracheal intubation	AUC (95% CI)	Cutoff	Sensitivity (95% CI)	Specificity (95% CI)	PPV (95% CI)	NPV (95% CI)
BAR	0.870 (0.837–0.899)	>5.20	92.86 (66.1–99.8)	73.50 (69.3–77.4)	9.5 (7.8–11.4)	99.7 (98.1–100)
NLR	0.761 (0.720–0.798)	>15.43	78.57 (49.2–95.3)	70.94 (66.6–75.0)	7.5 (5.6–9.9)	99.1 (97.6–99.7)
CRP	0.739 (0.698–0.778)	>5.92	78.57 (49.2–95.3)	66.67 (62.2–70.9)	6.6 (5.0–8.7)	99.0 (97.4–99.6)
TG2018	0.854 (0.819–0.884)	>2	92.86 (66.1–99.8)	79.70 (75.8–83.3)	12.0 (9.8–14.7)	99.7 (98.3–100)

Abbreviations: AUC; area under the curve, CI; confidence interval, PTBD; percutaneous transhepatic bile drainage, BAR; blood urea nitrogen-to-albumin ratio, NLR; neutrophil-to-lymphocyte ratio, CRP; c-reactive protein, TG2018; Tokyo guideline 2018.

## Data Availability

The data used to support the findings of this study are available from the corresponding author upon request.

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
