# Peer review of "Clinical Utility of the Tokyo Guidelines 2018 for Acute Cholangitis in the Emergency Department and Comparison with Novel Markers (Neutrophil-to-Lymphocyte and Blood Nitrogen Urea-to-Albumin Ratios)"

_jcm, 2024, doi:10.3390/jcm13082306_

Round 1

Reviewer 1 Report

Comments and Suggestions for Authors

Overall, this is the first study to utilize TG2018 guideline for predicting consequences of AC in ED in which biomarkers such as NLR, BAR were used.

the study was novel and could attract wide readers in the field. However, some minute details need to be amended.

The following details need to be clarified or amended. (comments and line no. had been mentioned following the PDF manuscript format)

1. line no. 23, the full term of AC and ED needs to be stated here. Although the readers could guess the full name of AC and ED from the title of the study or read the first sentence of the introduction section. From my point of view,  full name of them should be added here.

2. line no.32-33, it is fine to skip the exclusion crtiteria on the abstract section, as readers can have a look at these details within the methods section if needed. So, exclusion criteria can be deleted here.

3.line no 106., the word "ratio" should be added for the phrase" prothrombin time-international normalized'', so this should be" prothrombin time-international normalized ratio"

4. line no.159, on table 1, should hemoglobin data be written as hemoglobin level or concentration? Although, in clinical setting, most readers from clinical fields could guess that this was the concentration data because it was shown in a unit of g/dL. From my point of view, better to clarify it as concentration not types of them or others.

5.line no. 171, table 2, the overall table does not look fine. It should be adjusted in terms of positions of numbers or variables.  Feel free to make this table easy to read and follow like table no.3 (although the numbers of columns are not the same).

6.line no. 164, "long-term hospital days" should be re-written as "long-term hospital stays"

7.line no.192. "hemoglobin" should be re-written as "hemoglobin level".

8.line no.213, full term of AUC should be " area under the curve" as the authors first mentioned this parameter as per line no. 143.

Thank you for considering suggestions. 

Comments on the Quality of English Language

The overall English quality is fine and easy to read and follow.

Author Response

We sincerely thank the Editor and reviewers for their thoughtful recommendations for improving the quality of our manuscript ("Clinical utility of the Tokyo Guidelines 2018 for acute cholangitis in the emergency department and comparison with novel markers (neutrophil-to-lymphocyte and blood nitrogen urea to albumin ratios"). We have revised the manuscript based on the comments and suggestions of the reviewers. We have also provided point-by-point responses to the reviewers’ comments, and the changes made according to the reviewer’s suggestions are shown in red highlight in the enclosed revised manuscript.

Reviewer 1

Overall, this is the first study to utilize TG2018 guideline for predicting consequences of AC in ED in which biomarkers such as NLR, BAR were used.

the study was novel and could attract wide readers in the field. However, some minute details need to be amended.

The following details need to be clarified or amended. (comments and line no. had been mentioned following the PDF manuscript format)

  1. line no. 23, the full term of AC and ED needs to be stated here. Although the readers could guess the full name of AC and ED from the title of the study or read the first sentence of the introduction section. From my point of view, full name of them should be added here.

(Our response) In consideration of your comments, we revised.

  1. line no.32-33, it is fine to skip the exclusion crtiteria on the abstract section, as readers can have a look at these details within the methods section if needed. So, exclusion criteria can be deleted here.

(Our response) In consideration of your comments, we revised.

3.line no 106., the word "ratio" should be added for the phrase" prothrombin time-international normalized'', so this should be" prothrombin time-international normalized ratio"

(Our response) In consideration of your comments, we revised.

  1. line no.159, on table 1, should hemoglobin data be written as hemoglobin level or concentration? Although, in clinical setting, most readers from clinical fields could guess that this was the concentration data because it was shown in a unit of g/dL. From my point of view, better to clarify it as concentration not types of them or others.

(Our response) Sorry for confusion. In consideration of your comments, we revised.

5.line no. 171, table 2, the overall table does not look fine. It should be adjusted in terms of positions of numbers or variables.  Feel free to make this table easy to read and follow like table no.3 (although the numbers of columns are not the same).

(Our response) In consideration of your comments, we revised.

6.line no. 164, "long-term hospital days" should be re-written as "long-term hospital stays"

(Our response) In consideration of your comments, we revised.

7.line no.192. "hemoglobin" should be re-written as "hemoglobin level".

(Our response) In consideration of your comments, we revised.

8.line no.213, full term of AUC should be " area under the curve" as the authors first mentioned this parameter as per line no. 143.

(Our response) In consideration of your comments, we revised.

Thank you for considering suggestions.

Reviewer 2 Report

Comments and Suggestions for Authors

Review on the manuscript entitled: “Clinical utility of the Tokyo Guidelines 2018 for acute cholangitis in the emergency department and comparison with novel markers (neutrophil-to-lymphocyte and blood nitrogen urea to albumin ratios)”

Article addresses widely known classification and management algorithm for AC (Tokyo 2018 guidelines) and how TG18 in comparison with novel inflammatory markers can predict poor outcomes for patients with AC.

Grade III AC according to TG18 already means that patient is having organ dysfunction thus admission to ICU with organ support and early intervention is advocated, also patients with Grade III AC have poor outcomes comparing to Grade II and I patients. Grade II AC are at a high risk for developing organ dysfunction and associated complications, Grade I are relatively easy manageable. TG18 allows clinitians to triage patients and allocate resources, this is already widely known and accepted.

Introduction:

I can not agree with the statement (line 70): “In an ED-based setting, where efficiency and practicality are highly desired, clinicians 70 may find the application of various parameters challenging”. Application of TG18 in ED setting is very easy and does not require multiple parameters, online calculators are available that can be easily uploaded to smartphones.

The aim of the study is not clearly stated.

Methods:

35000 patients are emergency visits or hospital inpatients?

Primary endpoint is stated, were there any secondary endpoints, please clarify.

Why only PTBD is considered as a predictor of severity of cholangitis, ERCP and percutaneous cholecystostomy is also used as a method for resolving biliary obstruction. All methods should be included in primary endpoint analysis, please clarify.

To define outcomes for Grade III, II and I cholangitis, outcomes should be evaluated in each of the groups separately.

Results:

CRP levels are surprisingly low, patients with Grade III AC usually have long standing infection in bile ducts that gradually lead to sepsis and severe sepsis, CRP levels in this group of patients are extremely high. Please comment on this.

Procalcitonin levels in correlations with CRP are good predictors for development of severe sepsis and poor outcomes, are you assessing procalcitonin levels in your hospital, please add.

“The analysis included 81 patients requiring intensive care, 111 requiring long-term 164 hospital days (≥14 days), 49 requiring PTBD during hospitalization, and 14 requiring ETI 165 during hospitalization.” How many patients in each arm (ICU, long hospital stay, PTBD etc.) had Grade III, II and I AC? Admission to ICU and long hospital stay can be associated not only to cholangitis but age and other underlying medical conditions. Please comment how other factors rather than cholangitis itself influenced outcomes.

Long hospital stay is not only associated to clinical condition but possibility to discharge patients to step-down or lower level hospitals, especially in the group of elderly people, this should be also commented or added to limitation part.

Comments on the Quality of English Language

Extensive English editing is needed.

Author Response

We sincerely thank the Editor and reviewers for their thoughtful recommendations for improving the quality of our manuscript ("Clinical utility of the Tokyo Guidelines 2018 for acute cholangitis in the emergency department and comparison with novel markers (neutrophil-to-lymphocyte and blood nitrogen urea to albumin ratios"). We have revised the manuscript based on the comments and suggestions of the reviewers. We have also provided point-by-point responses to the reviewers’ comments, and the changes made according to the reviewer’s suggestions are shown in red highlight in the enclosed revised manuscript.

Review on the manuscript entitled: “Clinical utility of the Tokyo Guidelines 2018 for acute cholangitis in the emergency department and comparison with novel markers (neutrophil-to-lymphocyte and blood nitrogen urea to albumin ratios)”

Article addresses widely known classification and management algorithm for AC (Tokyo 2018 guidelines) and how TG18 in comparison with novel inflammatory markers can predict poor outcomes for patients with AC.

Grade III AC according to TG18 already means that patient is having organ dysfunction thus admission to ICU with organ support and early intervention is advocated, also patients with Grade III AC have poor outcomes comparing to Grade II and I patients. Grade II AC are at a high risk for developing organ dysfunction and associated complications, Grade I are relatively easy manageable. TG18 allows clinitians to triage patients and allocate resources, this is already widely known and accepted.

Introduction:

  1. I can not agree with the statement (line 70): “In an ED-based setting, where efficiency and practicality are highly desired, clinicians 70 may find the application of various parameters challenging”. Application of TG18 in ED setting is very easy and does not require multiple parameters, online calculators are available that can be easily uploaded to smartphones.

The aim of the study is not clearly stated.

(Our response)

We agree with your comment. However, the purpose of this study was to compare TG2018 with novel markers (NLR, BAR) that can be represented by one or two parameters.

In addition, the purpose of this study is to address the question of whether TG2018 with parameters such as Part A (systemic inflammation - Fever, WBC count, CRP level), Part B (cholestasis - Total bilirubin level, abnormal liver enzymes), and Part C (Biliary dilatation, evidence of the etiology on imaging) can predict various clinical outcomes in the emergency department.

The TG2018 score is calculated based on the physiologic criteria and imaging tests described above. Therefore, we thought that it would be better for clinicians to be able to predict the outcome by simply looking at the levels of neutrophil, lymphocyte, BUN, and albumin before these results are available.

Methods:

  1. 35000 patients are emergency visits or hospital inpatients?

(Our response)

We apologize for the confusion. It is 35000 emergency visits.

  1. Primary endpoint is stated, were there any secondary endpoints, please clarify.

(Our response)

We apologized for the confusion. We revised it as follow

The secondary endpoint was to evaluate whether NLR, BAR measured in the emergency department are predictors of clinical outcomes. Predictive performance was compared to TG2018.

  1. Why only PTBD is considered as a predictor of severity of cholangitis, ERCP and percutaneous cholecystostomy is also used as a method for resolving biliary obstruction. All methods should be included in primary endpoint analysis, please clarify.

(Our response)

We have reanalyzed the data according to your suggestion and presented it in the supplementary table as follow. ERCP and TG2018 did not show a statistically significant difference. PTGBD is used for acute cholecystitis and PTBD was performed for the treatment of acute cholangitis in this patient group.

325 patients underwent ERCP, and the TG2018 and ERCP was not statistically significant (Supplemental Table 1).

Supplement table 1.

Factor

ERCP

p

No (n=155)

Yes (n=325)

BAR

4.09 (1.12-111.11)

3.73 (1.00-24.37)

0.006

NLR

10.04 (0.95-194.20)

9.59 (0.67-88.45)

0.866

CRP

4.72 (0.04-26.15)

2.92 (0.03-31.02)

0.035

TG2018

2 (1-3)

1 (1-3)

0.102

Abbreviations: ERCP; Endoscopic retrograde choangiopancreatography, BAR; blood urea nitrogen to albumin ratio, NLR; neutrophil to lymphocyte ratio, CRP; c-reactive protein, TG2018; tokyo guideline 2018,.

  1. To define outcomes for Grade III, II and I cholangitis, outcomes should be evaluated in each of the groups separately.

(Our response)

As your consideration, we revised it and represented supplementary table 2 as follows.

The results for each outcome by TG2018 grade are presented in supplementary table 2.

Supplement table 2.

Grade 1 (n=254)

Grade 2 (n=120)

Grade 3 (n=108)

P

Intensive care

8 (3.1%)

12 (10.0%)

61 (56.5%)

0.000

Long-term hospital stays

33 (13.0%)

30 (25.0%)

48 (44.4%)

0.000

PTBD

17 (6.7%)

9 (7.5%)

23 (21.3%)

0.000

Endotracheal intubation

1 (0.4%)

0

13 (12.0%)

0.000

Abbreviations: PTBD; percutaneous transhepatic bile drainage

Results:

  1. CRP levels are surprisingly low, patients with Grade III AC usually have long standing infection in bile ducts that gradually lead to sepsis and severe sepsis, CRP levels in this group of patients are extremely high. Please comment on this.

(Our response)

As your consideration, we revised supplementary table. We added it as follows.

BAR, NLR, and CRP levels by TG2018 grade were presented, and there were statistically significant differences as the severity of AC increased. (Supplementary table 3)

Supplement table 3.

Grade 1 (n=254)

Grade 2 (n=120)

Grade 3 (n=108)

P

BAR

3.24 (1.00-24.37)

4.33 (1.47-23.82)

6.34 (1.23-111.11)

0.000

NLR

6.74 (0.76-79.92)

12.74 (0.67-81.33)

15.66 (0.81-194.20)

0.000

CRP

1.16 (0.03-26.15)

5.61 (0.11-30.14)

7.53 (0.03-31.02)

0.000

Abbreviations: BAR; blood urea nitrogen to albumin ratio, NLR; neutrophil to lymphocyte ratio, CRP; c-reactive protein,

  1. Procalcitonin levels in correlations with CRP are good predictors for development of severe sepsis and poor outcomes, are you assessing procalcitonin levels in your hospital, please add.

(Our response)

Unfortunately, we did not include this blood test in this study.

This is because PCT is a useful blood test, but it is often not covered by insurance when performed in the emergency department.

8.“The analysis included 81 patients requiring intensive care, 111 requiring long-term hospital days (≥14 days), 49 requiring PTBD during hospitalization, and 14 requiring ETI during hospitalization.” How many patients in each arm (ICU, long hospital stay, PTBD etc.) had Grade III, II and I AC?

(Our response)

As your consideration, we revised it and represented supplementary table 2 as follows.

The results for each outcome by TG2018 grade are presented in supplementary table 2.

Supplement table 2.

Grade 1 (n=254)

Grade 2 (n=120)

Grade 3 (n=108)

P

Intensive care

8 (3.1%)

12 (10.0%)

61 (56.5%)

0.000

Long-term hospital stays

33 (13.0%)

30 (25.0%)

48 (44.4%)

0.000

PTBD

17 (6.7%)

9 (7.5%)

23 (21.3%)

0.000

Endotracheal intubation

1 (0.4%)

0

13 (12.0%)

0.000

Abbreviations: PTBD; percutaneous transhepatic bile drainage

  1. Admission to ICU and long hospital stay can be associated not only to cholangitis but age and other underlying medical conditions. Please comment how other factors rather than cholangitis itself influenced outcomes.

Long hospital stay is not only associated to clinical condition but possibility to discharge patients to step-down or lower level hospitals, especially in the group of elderly people, this should be also commented or added to limitation part.

(Our response)

As your consideration, we revised limitation as follows.

Due to the retrospective nature of the study, there are many factors that may contribute to increased length of hospital stay, and we cannot rule out confounding by all of them. In particular, intensive care and long-term hospital stay can be associated not only to cholangitis but age and other underlying medical conditions. Long-term hospital stay is not only associated to clinical condition but possibility to discharge patients to step-down or lower-level hospitals, especially in the group of elderly people.

We're attaching the English translation certificate. Thank you. 

Round 2

Reviewer 2 Report

Comments and Suggestions for Authors

Authors have made substential improvements to the manuscript. I have no further comments.

Author Response

Thank you for your good comments. Thanks to the reviewer, we were able to make good revision.